ated in many cases

# Neuropeptide F signaling regulates parasitoid-specific germline development and egg-laying in *Drosophila*

**Madhumala K. Sadanandappa**[1], **Shivaprasad H. Sathyanarayana**[1], **Shu Kondo**[2], **Giovanni Bosco**[1]*

**1** Department of Molecular and Systems Biology, Geisel School of Medicine at Dartmouth, Hanover, New Hampshire, United States of America, **2** Invertebrate Genetics Laboratory, National Institute of Genetics, Mishima, Shizuoka, Japan

* Giovanni.Bosco@dartmouth.edu

**Data Availability Statement:** All relevant data are within the manuscript and its Supporting Information files.

## Abstract

*Drosophila* larvae and pupae are at high risk of parasitoid infection in nature. To circumvent parasitic stress, fruit flies have developed various survival strategies, including cellular and behavioral defenses. We show that adult *Drosophila* females exposed to the parasitic wasps, *Leptopilina boulardi*, decrease their total egg-lay by deploying at least two strategies: Retention of fully developed follicles reduces the number of eggs laid, while induction of caspase-mediated apoptosis eliminates the vitellogenic follicles. These reproductive defense strategies require both visual and olfactory cues, but not the *MB247*-positive mushroom body neuronal function, suggesting a novel mode of sensory integration mediates reduced egg-laying in the presence of a parasitoid. We further show that neuropeptide F (NPF) signaling is necessary for both retaining matured follicles and activating apoptosis in vitellogenic follicles. Whereas previous studies have found that gut-derived NPF controls germ stem cell proliferation, we show that sensory-induced changes in germ cell development specifically require brain-derived NPF signaling, which recruits a subset of NPFR-expressing cell-types that control follicle development and retention. Importantly, we found that reduced egg-lay behavior is specific to parasitic wasps that infect the developing *Drosophila* larvae, but not the pupae. Our findings demonstrate that female fruit flies use multimodal sensory integration and neuroendocrine signaling via NPF to engage in parasite-specific cellular and behavioral survival strategies.

## Author summary

Behavioral adaptation to environmental threats such as infectious diseases or predators increases the survival and fitness of an organism. Here, we studied behavioral immunity in adult *Drosophila* females that protect their progeny from the parasitic infection. We show that *Drosophila* females modify their oviposition behavior in the presence of a parasitic wasp. This change in reproductive behavior is highly specific to *Leptopilina* wasps, which necessitates both visual as well as the parasitoid-specific olfactory cues. In addition,

**Funding:** This study was supported by the following grants: MKS: Human Frontier Science Program Long-Term Fellowship. https://www.hfsp.org/funding/hfsp-funding/postdoctoral-fellowships GB: National Institute of Health, Pioneer grant DP1MH110234. https://commonfund.nih.gov/pioneer. The funders had no role in study design, data collection and analysis, decision to publish, or preparation of the manuscript.

**Competing interests:** The authors have declared that no competing interests exist.

we find that the transient retention of matured follicles and increased apoptosis of the developing follicles in the parasitoid-exposed *Drosophila* ovaries results in an egg-lay reduction. We also identify that the neuroendocrine signaling involving neuropeptide F (NPF) and its cognate receptor, NPFR, mediates the parasitoid-induced egg-lay depression and germline physiological modifications. Based on the innate recognition of the predatory threat, our study unravels the cellular and physiological mechanisms that underlie an ecological relevant form of behavioral adaptation in *Drosophila*.

## Introduction

Organisms have developed various survival strategies to circumvent the strong selection pressure imposed by environmental threats. Parasitism is one such threat that is ubiquitous throughout all levels of biology, and this pressure has given rise to a myriad of both general and highly specific protective strategies to thwart parasitism. For instance, *Melanoplus sanguinipes*, a migratory grasshopper, prefers high temperature and displays thermoregulation to prevent fungal parasite infection [1]. Similarly, gypsy moth (*Lymantris dispar*) larvae detect and avoid virus-infected cadavers [2], pea aphids (*Acyrthosiphon pisum*) show bacterial endosymbiosis to protect from aphid-specific fungal pathogens [3], and woolly bear caterpillars self-medicate plant toxins—pyrrolizidine alkaloids as a defense against tachinid fly endoparasitoids [4]. Interestingly, some of these behavioral adaptations that protect against potentially lethal threats are also beneficial to an individual's subsequent generations. For example, a most recent study has shown that parasitoid exposure leads to transgenerational inheritance of ethanol-seeking behavior in *Drosophila* [5]. Likewise, a parasitoid-infected adult monarch butterfly preferentially oviposits on the medicinal plant that reduces the parasitoid development and disease in their offspring caterpillars [6,7]. Despite the significance and ubiquity in insects, the neuronal circuit(s) and molecular mechanisms that drive various forms of insect behavioral immunity remain poorly understood.

In nature, up to 90% of *Drosophila* larvae are parasitized by different wasp species [8,9]. Among them, *Leptopilina boulardi* and *Leptopilina heterotoma* are the most common larval parasitoids that infect the developing fruit fly larvae. *L. boulardi* is a specialist parasite that mostly parasitizes *D. melanogaster* and *D. simulans* clade, whereas *L. heterotoma* is a generalist parasite that successfully infects diverse species of *Drosophila* [10]. If the parasitized fly larvae fail to encapsulate the eggs [11], then the eggs develop into parasitoid larvae that consume the fly larva before eclosing from *Drosophila* pupal case. On the other hand, to escape the parasitoid pressure, both *Drosophila* larvae, as well as adult flies, have developed various behavioral responses that reduce the risk of infection. For instance, *Drosophila* larvae exhibit a specific rolling behavior when attacked by a wasp, and this escape response is controlled by a multimodal circuit involving mechanosensory and nociceptive neurons [12,13]. Both larvae and adult flies display a parasitoid-avoidance behavior, which is mediated by a specific olfactory receptor neuron (ORN) that expresses the odorant receptors (OR), Or49a and Or85f, which selectively respond to *Leptopilina* odors. While the *Drosophila* larvae crawl away to escape the infection, the adult females avoid laying the eggs where the *Leptopilina* wasps are present [14]. Lastly, in the presence of a parasitic wasp, adult *Drosophila* females not only suppress their total egg-lay [15,16] but also prefer to oviposit in food containing toxic levels of alcohol [17,18]. This preferential alteration in ethanol-seeking behavior self-medicates the larvae against wasp infection, thereby prevents wasp adults emerging from fly pupae [17].

Given that *Leptopilina* wasps attack only the fly larvae, the observed parasitoid-triggered behavioral adaptation in adult fruit flies is presumably to decrease the number of progeny vulnerable to infection. It is interesting to note that these behavioral responses are independent of prior experiences (i.e. innate), and the laboratory strains removed from such natural pressures for hundreds of generations still exhibit robust responses to wasp exposure. As such, this innate behavior likely represents an early-evolved neuronal circuit that must be integrated into other essential processes that have persisted in the absence of natural predator pressures. However, how adult *Drosophila* specifically recognizes different parasitoids, for example, *Leptopilina*, and modulate their innate responses in a parasitoid-specific manner remains an open question. Moreover, how germline cells receive signals in response to parasitoid exposure is unclear. To address these questions, we investigated the underlying mechanisms of parasitoid-induced egg-lay reduction in *Drosophila* females in terms of (i) the required sensory modalities, (ii) the downstream circuits that they recruit, and (iii) the subsequent germline modifications that account for the altered reproductive behavior. Our results suggest that the decreased egg-lay is not a generic stress response of adult females to all parasitoid wasps. We show that brain-derived *Drosophila* Neuropeptide F (NPF) signaling requires both visual and olfactory inputs to modify germline physiology and oviposition behavior in response to *Leptopilina* parasitoids. We provide evidence for innate recognition of predatory threat by adult fruit flies that can distinguish between different parasitoid species.

## Results

### *Drosophila* females depress their total egg-lay upon exposure to *Leptopilina* wasp

To test whether the parasitoid-induced alteration in reproductive behavior is a general stress response or a wasp-specific behavioral modification, we examined the egg-lay responses of wildtype Canton S (CS) flies to different species of parasitoid wasps. *Leptopilina boulardi* (Family Figitidae; strain Lb17) is a larval parasitoid wasp that infects the developing fruit fly larvae [9]. Twenty-four hrs of exposure to either female or male Lb17 parasitoids elicits a significant reduction in mean egg-lay of CS females compared to their mock controls (wasp-exposed–Fig 1A and 1B and Table 1). Consistent with the egg-lay reduction, the wasp-exposed group also showed the same proportional decrement in the mean eclosion than unexposed controls (wasp-exposed - S1A Fig). Remarkably, the percentage of CS egg-lay and the average number of flies eclosed from mock and exposed groups are comparable 24 hrs before wasp exposure (pre-exposed–Figs 1B and S1A and Table 1) and 24 hrs after wasp removal (post-exposed–Figs 1B and S1A and Table 1). This finding suggests that the observed Lb17-induced egg-lay reduction is not due to a deficiency or an inability of *Drosophila* females to lay eggs (Mean egg-lay responses for total 72 hrs including, all three time intervals (pre-exposed, wasp-exposed, and post-exposed): 280.75 ± 11.20 for mock vs. 255.5 ± 3.89 for Lb17 ♀-exposed, $p = 0.05$; vs. 246.67 ± 6.88 for Lb17 ♂-exposed, $p = 0.018$).

A 24 hrs-exposure to female pupal parasitoids, such as *Trichopria sp.1* (Family Diapriidae; strain Trical) and *Pachycrepoideus sp.1* (Family Pteromalidae; strain Pac1Port) [19], which exclusively infect *Drosophila* pupae, fails to trigger an egg-lay decrease in wasp-exposed CS flies (Fig 1C and Table 1). However, to rule out the possibility that the wildtype CS strain is unable to perceive the pupal parasitoids, we investigated the wasp-induced egg-lay behavior in a different *Drosophila* wildtype strain, Oregon R (OR). Similar to CS flies, OR females also decreased their total egg-lay upon exposure to Lb17 parasitoids, but not to pupal wasps (S1B and S1C Fig and Table 1).

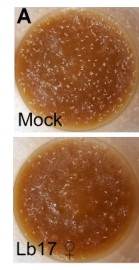

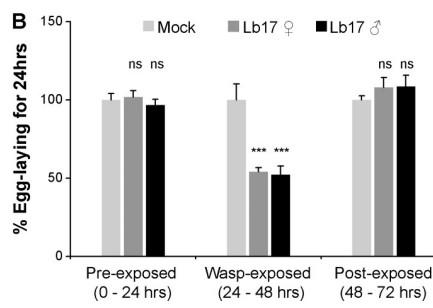

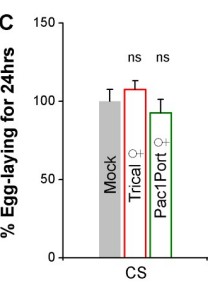

**Fig 1. Larval parasitoids induce egg-lay depression in *Drosophila* females.** (**A**) Representative image showing 24 hrs egg-lay of mock and Lb17 ♀-exposed flies. (**B**) Histogram showing the percent egg-laying responses of wildtype CS fruit flies (wasp-exposed) to Lb17 female (♀ - grey bars) and Lb17 male wasps (♂ - black bars). Light grey bars correspond to egg-lay responses of controls that are devoid of parasitoids (mock). Pre-exposed and post-exposed bars respectively correspond to 24 hrs mean egg-lay responses of CS females before and after Lb17-exposure. The egg-lay responses represented as a percentage of the mock response of the flies (% egg-laying for 24 hrs). (**C**) Oviposition behavior of CS flies in the presence of female pupal parasitoids–Trical (red) and Pac1Port (green). Error bars are ± SEM. *** $p \leq 0.001$ and ns for non-significance ($p > 0.05$) calculated using Student's *t*-test. For **B** and **C**: shown is the average egg-lay responses ± SEM. Refer to **Table 1** for 'N' and $p$ values.

## Retention of matured follicles and apoptosis of vitellogenic follicles leads to egg-lay depression

To identify the causes of Lb17-induced egg-lay depression, we asked whether the wasp-exposure had any physiological consequences in germline development. In order to address this, we dissected the ovaries from mock and wasp-exposed *Drosophila* females and systematically quantified the number of follicles at different developmental stages, the number of apoptotic egg chambers, and the number of germ stem cells (GSCs) (Figs 2 and S2).

An earlier study has shown that exposure to *Leptopilina heterotoma* (Family Figitidae; strain Lh14), a larval parasitic wasp, elicits reduced egg-lay as a consequence of apoptosis of the developing follicles, resulting in small ovaries [16]. Therefore, we immunolabeled the dissected ovaries for Dcp-1, an effector caspase in *Drosophila* required for nurse cell death during mid-oogenesis [20,21]. Similar to *L. heterotoma* (Lh14), 24 hrs of exposure to *L. boulardi* (Lb17) also increased the number of apoptotic egg chambers in the *Drosophila* female germline (Figs 2A, 2B and S2A). Despite an increased cell death of vitellogenic follicles, we noticed larger ovaries in Lb17-exposed females compared to unexposed controls (Figs 2D and S2A). Interestingly, the accumulation of matured stage 14 follicles accounts for enlarged ovaries in CS and OR females exposed to Lb17 wasps (Figs 2D, 2E, and S2). Additionally, we observed a reduced number of stages-10 to 13 egg chambers in Lb17-exposed ovaries (Figs 2E and S2B). Together, these findings suggest that in *Leptopilina*-exposed females, stages-10 to 13 follicles continue to develop and arrest at stage 14, while the elimination of vitellogenic follicles through apoptosis fails to replenish these intermediate developmental stages (S2A Fig).

Upon removal of the parasitoids, the mean egg-lay responses of 24 hrs Lb17-exposed CS females are comparable to unexposed controls (56.75 ± 1.56 for mock vs. 61.25 ± 3.61 for Lb17 ♀-exposed, $p = 0.271$; vs. 61.67 ± 4.05 for Lb17 ♂-exposed, $p = 0.276$) (post-exposed–Fig 1B) hints that Lb17-induced germline modifications are transient and reversible. To test, whether Lb17-exposure affects the GSC number and survivability, we next analyzed the germarium, which contains GSCs that generates one GSC for self-renewal and one cystoblast that differentiate to produce an oocyte [22]. The immunostaining analysis of GSCs number revealed no significant differences between unexposed and wasp-exposed groups (Fig 2F), indicating that 24 hrs of *L. boulardi* exposure may not have a long-lasting effect on egg development.

**Table 1. Raw mean egg-lay\* of all genotypes in different experimental conditions.**

| Genotypes | Exposure | Pre-exposed (n) | Wasp-exposed (n) | Post-exposed (n) | p-value |
| --- | --- | --- | --- | --- | --- |
| CS | Mock | 152.92 ± 6.34 (12) | 62.75 ± 5.86 (12)* | 56.75 ± 1.56 (12) | |
| CS | Lb17 ♀ | 155.75 ± 6.22 (12) | 38.50 ± 1.89 (12)* | 61.25 ± 3.61 (12) | $p^{**} < 0.001$ |
| CS | Lb17 ♂ | 147.83 ± 5.78 (12) | 37.16 ± 3.87 (12)† | 61.67 ± 4.05 (12) | $p^{*\dagger} < 0.001$ |

| Genotypes | Exposure | Mock (n) | Wasp-exposed (n) | p-value |
| --- | --- | --- | --- | --- |
| CS | Trical ♀ | 97.12 ± 7.28 (25) | 104.4 ± 5.43 (25) | $p = 0.427$ |
| CS | Pac1Port ♀ | 97.12 ± 7.28 (25) | 89.96 ± 8.20 (25) | $p = 0.517$ |
| OR | Lb17 ♀ | 143.75 ± 3.76 (12) | 104.91 ± 11.69 (12) | $p < 0.001$ |
| OR | Pac1Port ♀ | 143.75 ± 3.76 (12) | 138.58 ± 5.31 (12) | $p = 0.437$ |
| OR | Trical ♀ | 70.08 ± 8.51 (12) | 75.33 ± 4.07 (12) | $p = 0.774$ |
| CS–DD | Lb17 ♀ | 171.13 ± 7.52 (15) | 176.27 ± 6.94 (12) | $p = 0.620$ |
| CS | Lb17 ♀ | 194.92 ± 12.17 (12) | 131.5 ± 6.19 (12) | $p < 0.001$ |
| $ninaB^1$ | Lb17 ♀ | 183.33 ± 17.99 (12) | 164.42 ± 11.82 (12) | $p = 0.391$ |
| CS > UAS-Kir2.1 | Lb17 ♀ | 199.9 ± 14.33 (10) | 114.7 ± 19.25 (10) | $p = 0.003$ |
| ey-GAL4 > UAS-Kir2.1 | Lb17 ♀ | 195.33 ± 7.51 (15) | 199.53 ± 5.75 (15) | $p = 0.660$ |
| CS | Lb17 ♀ | 168 ± 8.60 (24) | 114.88 ± 6.17 (24) | $p < 0.001$ |
| $Orco^1$ | Lb17 ♀ | 125.44 ± 3.75 (18) | 122.61 ± 4.70 (18) | $p = 0.641$ |
| CS > UAS-Or49a RNAi | Lb17 ♀ | 237.65 ± 5.14 (17) | 182.65 ± 11.68 (17) | $p < 0.001$ |
| Or49a-GAL4 > UAS-Or49a RNAi | Lb17 ♀ | 195.41 ± 8.57 (17) | 173.18 ± 11.03 (17) | $p = 0.122$ |
| CS > UAS-Or85f RNAi | Lb17 ♀ | 248.17 ± 5.17 (12) | 182.83 ± 7.11 (12) | $p < 0.001$ |
| Or85f-GAL4 > UAS-Or85f RNAi | Lb17 ♀ | 234.2 ± 15.49 (15) | 220.56 ± 13.16 (16) | $p = 0.508$ |
| CS > UAS-Or56a RNAi | Lb17 ♀ | 213.67 ± 11.89 (18) | 158.11 ± 9.18 (18) | $p < 0.001$ |
| Or56a-GAL4 > UAS-Or56a RNAi | Lb17 ♀ | 180.61 ± 7.67 (18) | 138 ± 8.34 (19) | $p < 0.001$ |
| MB247-GAL4 > UAS-TNTVIF | Lb17 ♀ | 245.36 ± 12.92 (25) | 188.28 ± 11.64 (25) | $p = 0.002$ |
| MB247-GAL4 > UAS-TNT | Lb17 ♀ | 190.96 ± 8.24 (25) | 147.12 ± 9.53 (25) | $p < 0.001$ |
| CS | Lb17 ♀ | 148.45 ± 5.70 (58) | 103.79 ± 4.51 (58) | $p < 0.001$ |
| yw | Lb17 ♀ | 96.6 ± 5.97 (15) | 59.67 ± 8.25 (15) | $p < 0.001$ |
| $NPF^{SK1}$ | Lb17 ♀ | 126.96 ± 10.34 (24) | 112.67 ± 7.14 (24) | $p = 0.262$ |
| $NPF^{SK2}$ | Lb17 ♀ | 228.33 ± 6.74 (24) | 232.88 ± 5.89 (24) | $p = 0.614$ |
| $NPFR^{SK8}$ | Lb17 ♀ | 126.96 ± 4.97 (24) | 126.92 ± 7.56 (24) | $p = 0.997$ |
| CS > UAS-NPF RNAi | Lb17 ♀ | 143.87 ± 10.46 (15) | 88.13 ± 9.54 (16) | $p < 0.001$ |
| 25681 (NPF-GAL4) > UAS-NPF RNAi | Lb17 ♀ | 199.65 ± 15.05 (17) | 177.47 ± 12.55 (17) | $p = 0.267$ |
| 25682 (NPF-GAL4) > UAS-NPF RNAi | Lb17 ♀ | 137.31 ± 10.30 (16) | 126.44 ± 8.22 (16) | $p = 0.416$ |
| CS > UAS-NPF RNAi | Lb17 ♀ | 196.63 ± 15.08 (24) | 144.13 ± 13.94 (24) | $p = 0.014$ |
| 25681; nSyb-GAL80 > UAS-NPF RNAi | Lb17 ♀ | 145.77 ± 12.24 (26) | 96.55 ± 13.10 (29) | $p < 0.001$ |
| CS > UAS-NPFR RNAi | Lb17 ♀ | 214.77 ± 6.75 (30) | 158.4 ± 8.42 (30) | $p < 0.001$ |
| nanos-GAL4 > UAS-NPFR RNAi | Lb17 ♀ | 179.5 ± 8.51 (12) | 122.58 ± 16.97 (12) | $p < 0.001$ |
| mat ∝-GAL4 > UAS-NPFR RNAi | Lb17 ♀ | 195.25 ± 5.84 (12) | 101.41 ± 17.97 (12) | $p < 0.001$ |
| cb16-GAL4 > UAS-NPFR RNAi | Lb17 ♀ | 206.92 ± 9.72 (12) | 163.67 ± 13.12 (12) | $p = 0.015$ |
| c306-GAL4 > UAS-NPFR RNAi | Lb17 ♀ | 202.58 ± 11.33 (12) | 130.25 ± 17.22 (12) | $p = 0.002$ |
| e22c-GAL4 > UAS-NPFR RNAi | Lb17 ♀ | 210.33 ± 8.94 (15) | 157.33 ± 10.49 (15) | $p < 0.001$ |
| bab1-GAL4 > UAS-NPFR RNAi | Lb17 ♀ | 235.7 ± 7.99 (10) | 164.2 ± 14.60 (10) | $p < 0.001$ |
| GMR13C06-GAL4 > UAS-NPFR RNAi | Lb17 ♀ | 197.92 ± 13.25 (12) | 160.42 ± 11.57 (12) | $p = 0.045$ |
| CS > Tdc2-GAL4 | Lb17 ♀ | 219.04 ± 10.95 (28) | 174.61 ± 10.09 (28) | $p = 0.004$ |
| Tdc2-GAL4 > UAS-NPFR RNAi | Lb17 ♀ | 253.57 ± 6.60 (28) | 210.32 ± 9.87 (28) | $p < 0.001$ |
| CS > ppk-GAL4 | Lb17 ♀ | 167.6 ± 12.76 (15) | 115.8 ± 12.13 (15) | $p < 0.001$ |
| ppk-GAL4 > UAS-NPFR RNAi | Lb17 ♀ | 147.88 ± 8.95 (16) | 59.38 ± 12.96 (16) | $p < 0.001$ |
| CS > pLB1-GAL4 | Lb17 ♀ | 214.5 ± 8.78 (18) | 163.28 ± 7.97 (18) | $p < 0.001$ |
| pLB1-GAL4 > UAS-NPFR RNAi | Lb17 ♀ | 219.22 ± 8.99 (18) | 156.94 ± 10.63 (18) | $p < 0.001$ |

(*Continued*)

**Table 1.** (Continued)

| | | | | |
|---|---|---|---|---|
| CS > *UAS-NPFR RNAi* | Lb17 ♀ | 144.4 ± 11.07 (10) | 84.6 ± 10.72 (10) | *p* < 0.001 |
| 89E07-GAL4 > *UAS-NPFR RNAi* | Lb17 ♀ | 220 ± 6.03 (10) | 134.4 ± 8.43 (10) | *p* < 0.001 |
| 58F03-GAL4 > *UAS-NPFR RNAi* | Lb17 ♀ | 193 ± 11.53 (9) | 119.22 ± 3.17 (9) | *p* < 0.001 |
| 75G12-GAL4 > *UAS-NPFR RNAi* | Lb17 ♀ | 213.3 ± 8.11 (10) | 152.4 ± 10.99 (10) | *p* < 0.001 |
| 38E07-GAL4 > *UAS-NPFR RNAi* | Lb17 ♀ | 188.5 ± 10.66 (10) | 120.9 ± 7.03 (10) | *p* < 0.001 |
| CS > 60E02-GAL4 | Lb17 ♀ | 223.63 ± 15.07 (16) | 148.27 ± 15.49 (15) | *p* < 0.001 |
| 60E02-GAL4 > *UAS-NPFR RNAi* | Lb17 ♀ | 221.87 ± 22.61 (15) | 160.25 ± 20.27 (16) | *p* = 0.051 |
| CS > 60G05-GAL4 | Lb17 ♀ | 212.07 ± 8.96 (29) | 168.32 ± 9.61 (28) | *p* = 0.002 |
| 60G05-GAL4 > *UAS-NPFR RNAi* | Lb17 ♀ | 178.83 ± 9.82 (29) | 162.79 ± 8.78 (29) | *p* = 0.229 |
| CS > 61H06-GAL4 | Lb17 ♀ | 216.06 ± 9.58 (16) | 175.19 ± 5.59 (16) | *p* = 0.001 |
| 61H06-GAL4 > *UAS-NPFR RNAi* | Lb17 ♀ | 216.06 ± 21.47 (16) | 123.44 ± 13.88 (16) | *p* = 0.001 |
| CS > 65C12-GAL4 | Lb17 ♀ | 205.15 ± 8.92 (20) | 162.9 ± 9.05 (20) | *p* = 0.002 |
| 65C12-GAL4 > *UAS-NPFR RNAi* | Lb17 ♀ | 253.25 ± 15.02 (20) | 182.68 ± 16.68 (19) | *p* = 0.003 |
| CS > 60E02-GAL4 | Lb17 ♀ | 238.36 ± 6.66 (11) | 186 ± 5.64 (11) | *p* < 0.001 |
| 60E02-GAL4 > *UAS-Kir2.1* | Lb17 ♀ | 240.85 ± 8.41 (13) | 193.69 ± 3.89 (13) | *p* < 0.001 |
| CS > 60G05-GAL4 | Lb17 ♀ | 228.07 ± 6.75 (15) | 174.93 ± 9.64 (15) | *p* < 0.001 |
| 60G05-GAL4 > *UAS-Kir2.1* | Lb17 ♀ | 198.83 ± 11.99 (18) | 183.67 ± 13.22 (18) | *p* = 0.401 |
| CS > 61H06-GAL4 | Lb17 ♀ | 205.2 ± 9.40 (10) | 143.2 ± 7.68 (10) | *p* < 0.001 |
| 61H06-GAL4 > *UAS-Kir2.1* | Lb17 ♀ | 207.3 ± 14.55 (10) | 147.8 ± 11.11 (10) | *p* = 0.005 |
| CS > 65C12-GAL4 | Lb17 ♀ | 232.4 ± 12.10 (10) | 154.1 ± 11.74 (10) | *p* < 0.001 |
| 65C12-GAL4 > *UAS-Kir2.1* | Lb17 ♀ | 189.7 ± 9.72 (10) | 120.9 ± 15.23 (10) | *p* = 0.002 |

*values are ± SEM; *n* is the number of sets for each condition.

As expected, the proportion of cleaved Dcp-1 positive egg chambers, the ovary size, as well as the number of developing follicles from stages-10 to 14 are indistinguishable between pupal parasitoid-exposed and mock controls (Fig 2C–2E). Though Pac1Port-exposed ovaries showed a significant increase in the number of stages-10 to 13 follicles and a reduction in the number of matured follicles, the total number of developing egg chambers remain comparable to unexposed controls (Fig 2E). Collectively, our data demonstrate that in the presence of Lb17 larval parasitoids, *Drosophila* females transiently retain stage 14 follicles and trigger the mid-oogenesis checkpoint to eliminate the developing egg chambers through apoptosis, resulting in egglay reduction.

## Wasp-induced egg-lay depression requires visual signals

Preferential alteration of egg-lay behavior and germline physiological modifications to larval parasitoids (Lb17), but not to pupal parasitoids (Trical and Pac1Port), suggests that the adult fruit flies can recognize and distinguish between their predators [17]. How different types of wasps are perceived and distinguished from one another remains unclear. While previous reports demonstrated that the visual inputs are necessary for *Drosophila* to respond to *L. heterotoma* [15,16], we performed behavioral assays in the dark to test the role of vision in *L. boulardi*-induced behavioral response. During Lb17 exposure, instead of the standard 12:12 hrs LD cycles, the experimental flies are shifted to the constant dark regimes (DD). The absence of visual inputs not only blocks egg-lay depression (Fig 3A and Table 1) but also inhibits the retention of matured follicles in Lb17-exposed ovaries (Fig 3B). As a consequence, the ovary size and the number of follicles in the Lb17-exposed females are indistinguishable from unexposed controls (Figs 3B and S3A).

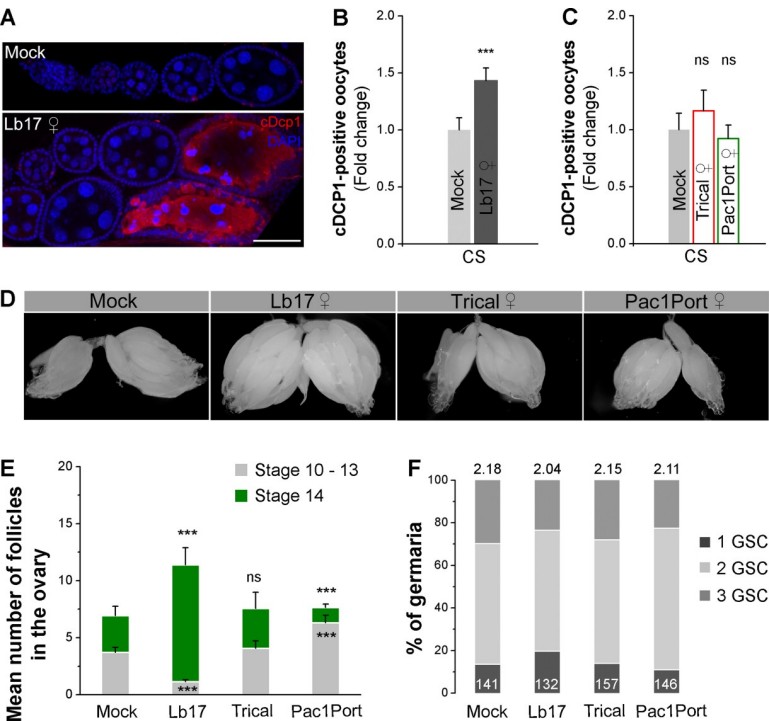

**Fig 2. Matured oocyte retention and increased apoptosis in the ovaries leads to egg-lay reduction.** (**A**) Confocal images of unexposed and Lb17 ♀-exposed fruit fly ovaries that are immunolabeled for DAPI (blue) and the apoptotic marker cleaved Dcp-1 (red). The scale bar corresponds to 50 μm. (**B** and **C**) Histogram showing the cDcp-1-positive follicles in CS ovaries that are either exposed to larval (Lb17) or pupal (Trical and Pac1Port) parasitoids. (**D**) Representative images of the ovaries dissected from mock and parasitoid-exposed CS files. (**E**) Stacked histogram showing the average number of stages-10 to 13 (light grey) and stage 14 (green) follicles in mock and parasitoid-exposed female ovaries. (**F**) The proportion of germaria containing one, two and three GSCs in mock and parasitoid-exposed females. The numbers on top of the bars correspond to the average number of GSCs per germarium, whereas the numbers on the bars (bottom) represent the total number of germaria analyzed. Error bars are ± SEM. *** $p \leq 0.001$ and ns for non-significance ($p > 0.05$) calculated using Student's *t*-test.

To further substantiate the above observation, we tested a specific *Drosophila* mutant that fails to perceive the visual information. A mutation in the *ninaB* gene, whose gene product is essential for visual pigment production [23,24], suppresses the Lb17-induced egg-lay responses. Compared to wildtype controls, Lb17-exposed *ninaB[1]* mutants neither declined their mean egg-lay nor retained the stage 14 follicles (Fig 3C–3E and Table 1). Similarly, inhibition of the photoreceptor neuronal activity by expression of an inward-rectifying potassium channel, Kir2.1 (*ey-GAL4 > UAS-Kir2.1*) [25], also reiterates behavioral phenotypes of *ninaB[1]* mutants (Fig 3D and Table 1). Consistent with the earlier studies [15], these findings confirm the requirement of visual cues in behavioral and germline modifications triggered by Lb17 parasitoids.

## ab10B neurons are responsible for *Leptopilina*-induced egg-lay depression

As mentioned before, both *Drosophila* larvae and adults avoid sites that smell like their predator, *Leptopilina* [14]. This innate avoidance response is conserved across several *Drosophila* species, mediated by an extremely-specific subset of ORNs that detect parasitoid-specific odors [14,26,27]. Given the importance of olfaction in *Leptopilina* avoidance behavior, we initially tested *Orco[1]* null mutants that lack a functional odorant co-receptor (Orco), which is expressed in most of the ORNs [28]. In the presence of Lb17 wasps, *Orco[1]* females failed to

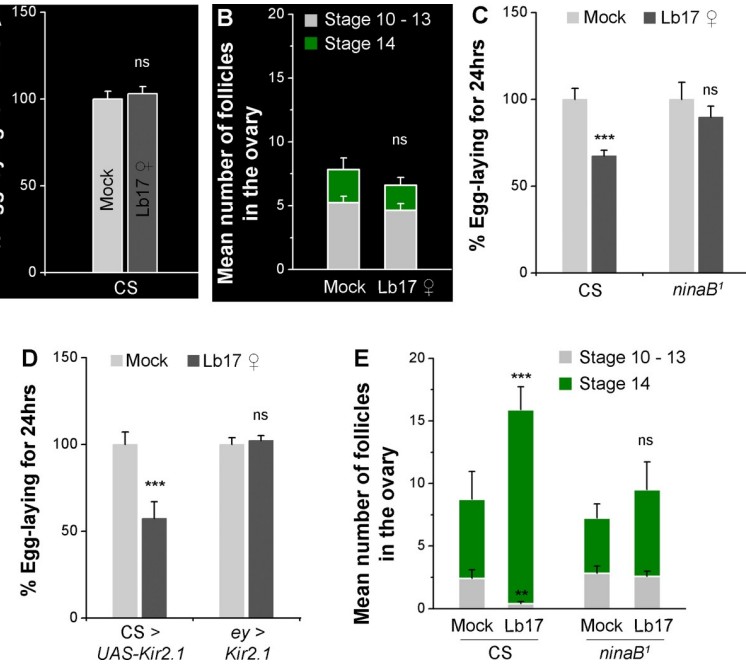

**Fig 3. Visual cues are necessary for Lb17-induced egg-lay decrease.** In the dark, Lb17 ♀-exposed CS flies fail (**A**) to depress their total egg-lay and (**B**) to transiently retain the stage 14 follicles in their ovaries. Lb17 ♀-induced egg-lay behavior in *ninaB¹* mutants (**C**) and flies expressing *UAS-Kir2.1* in photoreceptors (*ey-GAL4*) (**D**). (**E**) Stacked histogram showing the number of stages 10–13 (light grey), as well as stage 14 (green) follicles in mock and Lb17 ♀-exposed ovaries of CS and *ninaB¹* mutants. Error bars are ± SEM. ** $p \leq 0.01$, *** $p \leq 0.001$ and ns for non-significance ($p > 0.05$) calculated using Student's *t*-test. For **A, C** and **D**: mock and Lb17-exposed egg-lay responses are presented as light and dark grey bars, respectively along with mean ± SEM. Refer to **Table 1** for 'N' and *p* values.

display egg-lay reduction (Fig 4A and Table 1). Furthermore, knockdown of Orco receptors in ORNs (*Or83b-GAL4 > UAS-Orco RNAi*) (S3B Fig) and inhibition of the ORNs activity by overexpression of *UAS-Kir2.1* transgene [25] in the olfactory neurons (*Or83b-GAL4 > UAS-Kir2.1*) (S3C Fig), phenocopied the *Orco¹* mutant behavior for egg-laying. Blocking the synaptic neurotransmission from ORNs (*Or83b-GAL4 > UAS-TNTG*) [29,30] also failed to stall the stage 14 egg chambers in the Lb17-exposed female ovaries (Fig 4B). Together, these observations suggest that in addition to visual inputs, *Leptopilina*-specific olfactory cues are necessary for generating parasitoid-selective behavioral and germline modifications. In the absence of either a visual or olfactory sensory information, *Drosophila* females fail to alter their oviposition behavior in response to the parasitic pressure imposed by *L. boulardi* (Figs 3 and 4).

Previous studies have shown that a small basiconic sensillum of the adult *Drosophila* antenna (ab10B), which co-express two ORs (Or49a and Or85f) within the same ORN type, is necessary and sufficient to mediate avoidance behavior to the iridoid-producing *Leptopilina* [14,31]. Or49a receptor detects *Leptopilina* sex pheromone (-)-iridomyrmecin, whereas Or85f receptor responds to the parasitoid odors (R)-actinidine and nepetalactol [14,26,27,32–34]. Additionally, when activated alone, Or49a ORNs suppress oviposition in *Drosophila* females [35]. Therefore, to test the requirement of ab10B neurons in Lb17-induced egg-lay depression, we overexpressed OR-specific RNAi constructs in the ab10B ORNs. Knockdown of either Or49a or Or85f receptor in the ab10B neurons by using OR-specific GAL4 drivers (*Or49a-GAL4* and *Or85f-GAL4*) phenocopied the egg-lay behavior of *Orco¹* mutants (Fig 4C and 4D and Table 1), suggesting that the Lb17-induced egg-lay depression requires both olfactory receptors.

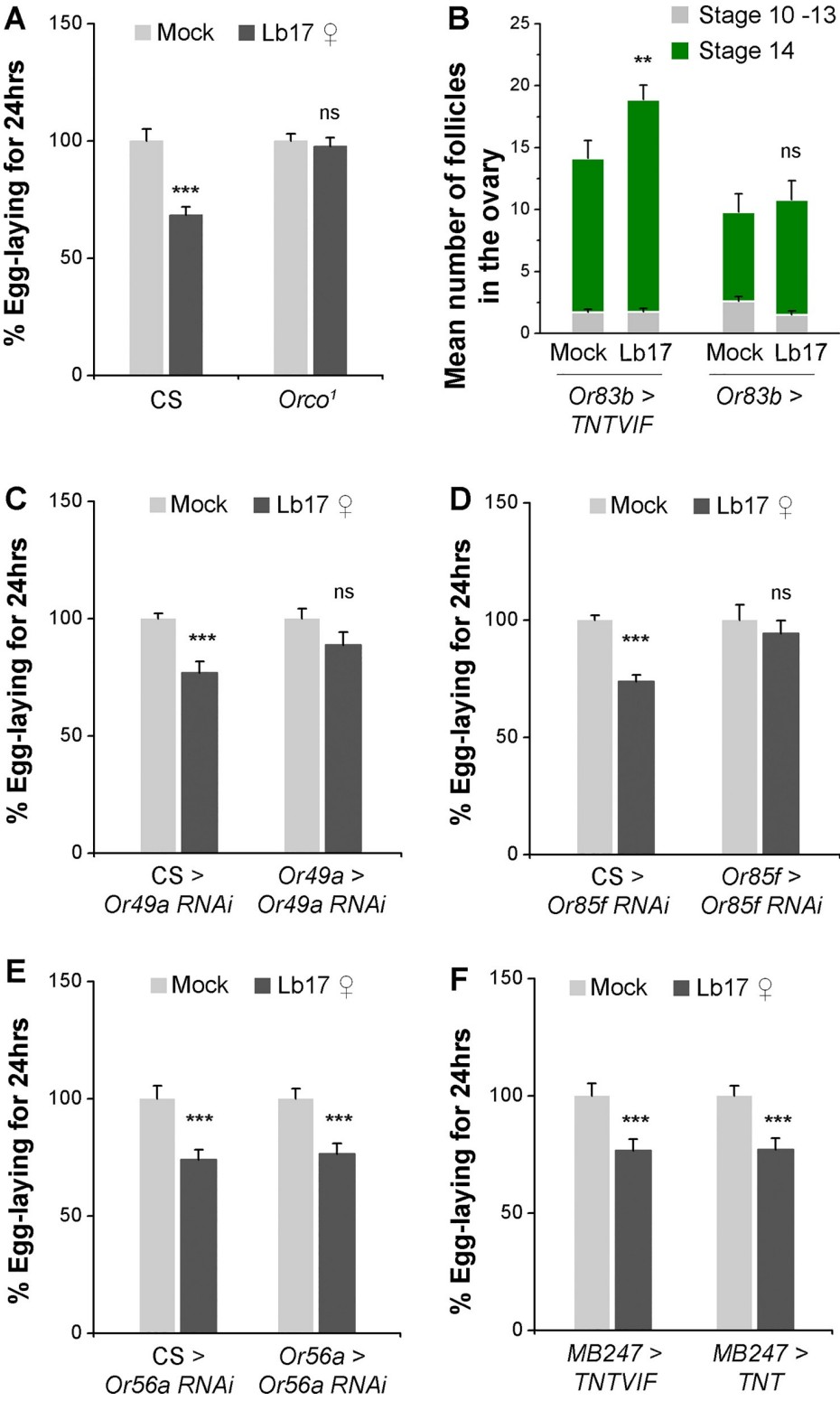

**Fig 4. The ab10B neurons selectively mediates Lb17-induced egg-lay depression.** (**A**) Histogram showing Lb17 ♀-induced egg-lay responses in *Orco¹* mutants. (**B**) Stacked histogram showing the average number of stages 10–13 (light grey) and stage 14 (green) follicles in mock and Lb17 ♀-exposed female ovaries expressing either *UAS-TNTVIF* (the inactive form of tetanus toxin light chain) or *UAS-TNT* (the active form) in ORNs. (**C-D**) Knockdown of either Or49a or Or85f receptor in the ab10B neurons blocks egg-lay depression in parasitoid-exposed females, (**E**) whereas,

Or56a knockdown does not affect the Lb17-induced egg-lay behavior. (**F**) Blocking synaptic neurotransmission from *MB247*-positive MB neurons has no significant effect on Lb-17 induced behavioral response. Error bars are ± SEM. $^{**}$ $p \leq 0.01$, $^{***}$ $p \leq 0.001$ and ns for non-significance ($p > 0.05$) calculated using Student's *t*-test. For **A**, **C**, **D**, **E**, and **F**: mock and Lb17-exposed egg-lay responses are presented as light and dark grey bars, respectively along with mean ± SEM. Refer to **Table 1** for 'N' and *p* values.

Given that the activation of the geosmin-specific olfactory circuit inhibits *Drosophila* egg-laying on food medium containing harmful microbes [36], we tested the role of geosmin-responding Or56a receptors expressed in the ab4B basiconic sensillum of the antenna [31] in wasp-induced egg-lay depression. Using previously validated GAL4-UAS constructs for Or56a receptors [36], we asked whether *Leptopilina*-triggered preferential alteration of egg-lay responses require these receptors. After 24 hrs of Lb17 exposure, *Or56a-GAL4 > UAS-Or56a RNAi* females showed a normal egg-lay reduction (Fig 4E and Table 1). We conclude that activation of the ab10B ORNs is necessary to trigger *Leptopilina*-selective behavioral and germline modifications, while Or56a receptors and their downstream circuits may not facilitate the observed egg-lay responses.

## *MB247*-positive mushroom body function is dispensable for egg-lay depression

The behavioral assays employed above measured an innate behavior in female *Drosophila* in the presence of wasps, requiring no previous exposure to a parasitic wasp. However, a learned behavioral response corresponds to the egg-lay reduction observed after wasp removal, involves a memory component [16]. A learned egg-lay depression to *L. heterotoma* mediated by a subset of mushroom body (MB) neurons [16], which are known to integrate and process different sensory modalities [37,38]. Though both visual (Fig 3) and olfactory (Fig 4) cues are required for *L. boulardi*-induced egg-lay depression, the comparable egg-lay responses of Lb17-exposed and mock controls upon wasp removal (Fig 1B–post-exposed) hints that the observed innate behavioral response in the presence of wasp is unlikely to be comprised of acquired memories. To test this hypothesis, we asked whether *Leptopilina*-induced innate behavioral response is MB-dependent by blocking the synaptic neurotransmission from *MB247*-positive MB neurons, whose function is implicated in learned egg-lay depression [16]. Expression of either an inactivated (*UAS-TNTVIF*) or an activated form (*UAS-TNT*) of the tetanus toxin light chain in *MB247*-positive MB neurons [29,30,39] failed to suppress the Lb17-induced behavioral response (Fig 4F and Table 1). This experiment thus suggests *MB247*-independent mechanisms in *L. boulradi*-generated innate behaviors.

## *NPF* and *NPFR* mutants fail to display a parasitoid-induced egg-lay reduction

In addition to regulating and coordinating various physiological and behavioral processes, neuropeptide signaling plays a crucial role in stress response [40–42]. For instance, exposure to female pheromones leads to increased *neuropeptide F* (*NPF*) mRNA and protein in the male fly brain that results in stress susceptibility and reduced life span. Male flies fail to respond to female pheromones after silencing *NPF*-expressing cells, suggest NPF role in transmitting the sensory information to the downstream neural circuits [43,44]. Similarly, in the presence of *L. heterotoma* wasps, visual cues-mediated reduction of NPF expression in the brain facilitates the ethanol-seeking behavior and caspase-mediated germline apoptosis in *Drosophila* females. While the preference for egg-laying in alcohol enriched media persisted even in the absence of a predator, how NPF triggers a cascade of behavioral and physiological modifications remains

elusive [5,17]. A recent study showed a mating-dependent increase in NPF expression in the midgut enteroendocrine cells (EECs) regulates germ stem cell proliferation in the ovaries [45]. Given the known function of NPF in both stress response and germline development, we examined the possible role of neuroendocrine signaling in Lb17-specific behavioral and germ-line modifications.

*Drosophila* NPF, an ortholog of mammalian neuropeptide Y (NPY), is a 36-residue ami-dated peptide that signals through a G protein-coupled receptor, the NPF receptor (NPFR) [40,41]. We tested previously described CRISPR/Cas9-generated *NPF* (*NPF*<sup>SK1</sup> and *NPF*<sup>SK2</sup>) and *NPFR* (*NPFR*<sup>SK8</sup>) null mutants for egg-lay behaviors [45,46]. Using a custom-made poly-clonal NPF antibody [5], we first confirmed the absence of peptide expression in *NPF* mutant fly brains (Fig 5A–5C). Upon exposure to *Leptopilina* wasp, both *NPF* mutants showed a block in egg-lay reduction (Fig 5D and Table 1). Likewise, mutant females deficient in NPF receptor activity (*NPFR*<sup>SK8</sup>) failed to decrease their mean egg-lay after 24 hrs of parasitoid exposure (Fig 5D and Table 1). Given CRISPR/Cas9-generated mutations are in an *yw* genetic background, we asked whether the lack of these two pigment genes may also interfere with the wasp percep-tion. We observed that *yw* control females with wild-type *NPF* and *NPFR* responded to Lb17 wasps by depressing their egg-lay to levels comparable to CS wild type controls (Fig 5D).

To confirm the behavioral phenotypes of *NPF* mutants, we first studied Lb17-triggered germline modifications in terms of matured follicles retention and stage-specific apoptotic events in the ovaries. As predicted, Lb17-exposed *NPF* mutants neither showed retention of the stage 14 egg chambers (Fig 5E) nor increased the number of cDcp-1-positive follicles in their ovaries (Fig 5F). Next, using previously validated *NPF-GAL4* drivers [47], we performed

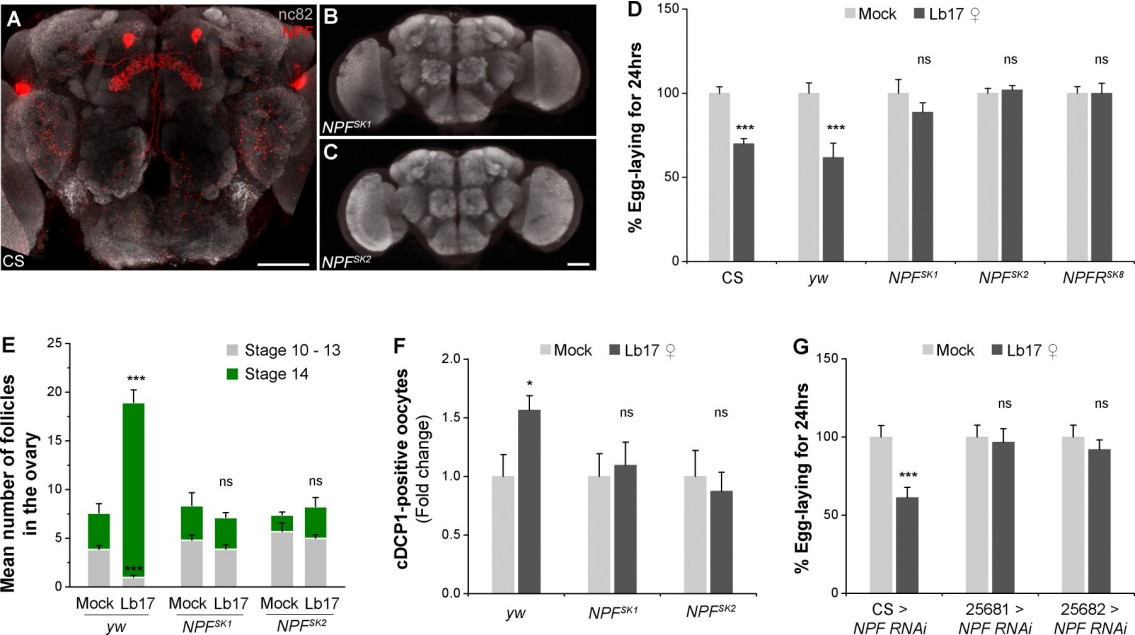

**Fig 5. *Drosophila NPF* and *NPFR* mutants are defective for parasitoid-induced egg-lay depression.** (**A-C**) Confocal projections of CS and *NPF* mutant fly brains immunolabeled for *NPF*-specific antibody (red) and bruchpilot (grey). The scale bar corresponds to 50 μm. (**D**) Histogram showing Lb17 ♀-induced egg-lay responses of *NPF* (*NPF*<sup>SK1</sup> and *NPF*<sup>SK2</sup>) and *NPFR* mutants (*NPFR*<sup>SK8</sup>). Compared to *yw* flies, (**E**) the number of stage 14 eggs and (**F**) the apoptotic vitellogenic follicles of parasitoid-exposed *NPF* mutants are indistinguishable from their mock controls. (**G**) Cell-specific NPF knockdown using two different *NPF-GAL4* drivers (25681 *and* 25682) prevents egg-lay depression after Lb17-exposure. Error bars are ± SEM. * $p \leq 0.05$, *** $p \leq 0.001$ and ns for non-significance ($p > 0.05$) calculated using Student's *t*-test. For **D** and **G**: mock and Lb17 ♀-exposed egg-lay responses are presented as light and dark grey bars, respectively, along with mean ± SEM. Refer to **Table 1** for 'N' and *p* values.

cell-specific RNAi-mediated knockdown experiments, which eliminated the anti-NPF signals from the central nervous system (CNS) (S4A and S4B Fig). The knockdown of NPF using two different *NPF-GAL4* drivers (# 25681 and # 25682) phenocopied the *NPF* mutant behavior for oviposition (Fig 5G and Table 1). Together, these findings confirmed the requirement of *NPF-NPFR* signaling in parasitoid-induced egg-lay depression.

## Egg-lay depression necessitates brain-derived NPF

In wildtype controls, we confirmed the presence of anti-NPF signals both in the CNS (Fig 5A) [47–49], as well as in a subset of enteroendocrine cells (EECs) of the midgut (Fig 6A) [45,50,51]. Midgut-derived NPF modulates mating-induced GSC proliferation in *Drosophila* germline [45,52]. Immunostaining analysis of *NPF* mutants and knockdown flies (*NPF-GAL4 > UAS-NPF RNAi*) revealed the absence of peptide expression both in the brain (Figs 5B, 5C and S4A') and midgut EECs (Fig 6A, 6B and 6B'). Consequently, it remained unclear whether the brain- or the midgut-expressed NPF is responsible for the observed egg-lay reduction after Lb17 exposure. Quantification of NPF-positive midgut EECs showed statistical insignificance between exposed and unexposed females (S4C Fig). Knockdown of NPF using the *Tachykinin-gut-GAL4* (*tk-gut-GAL4*) driver [51], expressed in a subset of NPF-positive EECs, leads to reduced basal egg-lay due to low levels of the midgut neuropeptide, as previously reported (S4D and S4F Fig) [45]. Interestingly, a recent report showed that *tk-gut-GAL4* expression is not specific to the mid-gut cells, it is also expressed in the fly brain [52]. Similarly, knockdown of NPF using a pan-neuronal GAL4 drivers such as *elav-GAL4* and *nSyb-GAL4* significantly altered the basal egg-lay relative to control flies (S4F Fig). Therefore, we performed the NPF knockdown analysis by co-expressing *NPF-GAL4* (# 25681) with the pan-neuronal GAL4 repressor, *nSyb-GAL80* [49,53]. In *NPF-GAL4; nSyb-GAL80 > UAS-NPF RNAi* transgenic flies, NPF expression is expected to be depleted in the midgut, but not in the CNS. In the EECs, lack of repressor expression leads to *NPF-GAL4*-mediated transgenic expression of RNAi against NPF, resulting in a drastic reduction of anti-NPF immunoreactivity in the midgut cells (Fig 6B). On the other hand, the CNS anti-NPF signal is detected in the cell bodies and the fan-shaped body (Fig 6C). Though, we observed reduced levels of NPF in the brain, the *NPF-GAL4; nSyb-GAL80 > UAS-NPF RNAi* females nonetheless displayed egg-lay reduction, retention of stage 14 follicles, and increased apoptosis in their ovaries upon 24 hrs of wasp exposure (Fig 6D–6G and Table 1). Our finding suggests that the midgut-derived NPF is dispensable for *Leptopilina*-induced egg-lay depression, unlike its function to promote mating-dependent GSC proliferation [45]. Additionally, we show that the neuronally-expressed NPF seems to mediate the Lb17-induced innate behavioral and physiological responses.

## *NPF-NPFR* signaling regulates parasitoid-specific alteration in reproductive behavior

We asked how does the CNS-derived NPF modulate the germline physiology to alter the egg-lay responses specific to *Leptopilina* wasp. In its downstream circuit, NPF binding to its cognate receptor, NPFR, activates the Gi intracellular signaling pathway to inhibit NPFR-expressing neurons [54]. To better understand the possible role of NPFR cells in Lb17-induced egg-lay depression, we performed cell-specific RNAi-mediated NPFR knockdown experiments. Transgenic expression of previously validated *UAS-NPFR RNAi* [45] in a different subset of ovarian cells such as the germ cells (*nos-GAL4* and *mat ∝-GAL4*), the somatic cells of ovarian germaria, including escort and follicle cells (*cb16-GAL4*, *e22c-GAL4*, and *13C06-GAL4*), the posterior follicle cells (*c306-GAL4*), and the cap cells (*bab1-GAL4*) [55]—all failed to block the

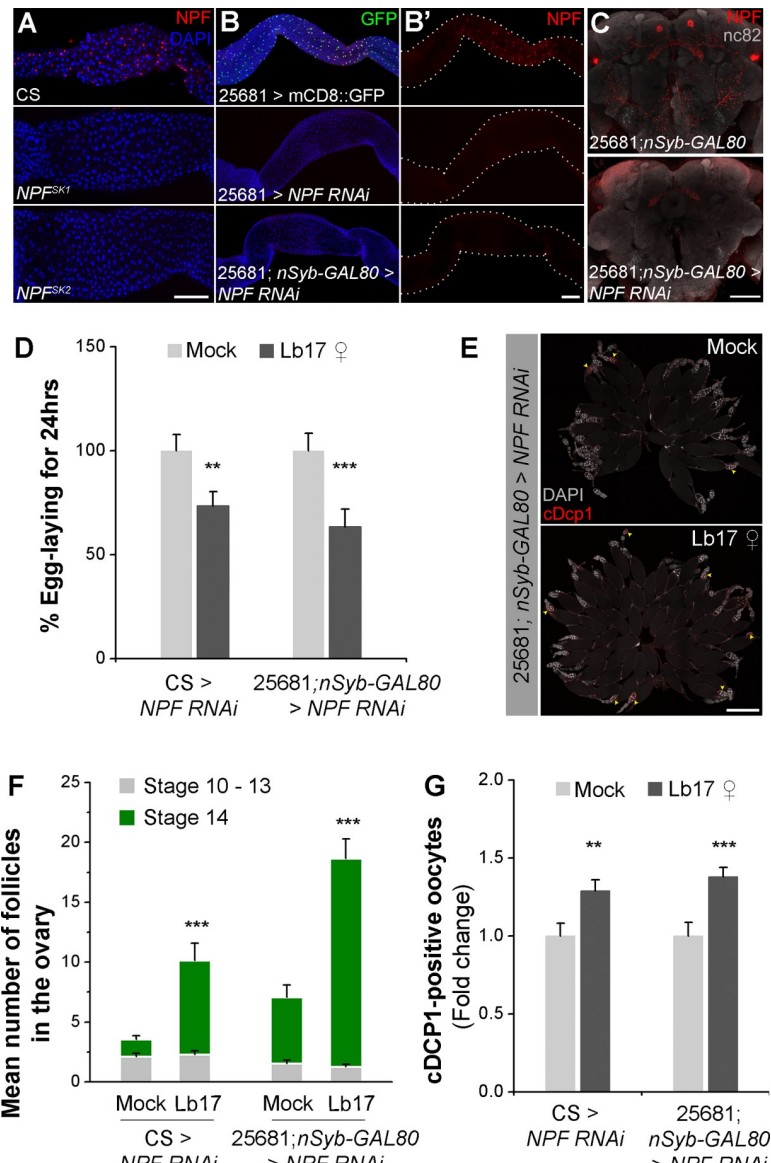

**Fig 6. Lb17-induced egg-lay depression requires CNS-derived NPF.** (**A**) Representative images of anti-NPF (red) immunostaining in the midgut EECs of CS and *NPF* mutants. (**B**) Flies expressing *UAS-NPF RNAi* under *NPF-GAL4* (25681) driver completely lacks anti-NPF (red) immunoreactivity in the middle midgut cells. For **A** and **B**: nuclei are stained with DAPI. The scale bar corresponds to 100 μm. The scale bar corresponds to 100 μm. (**C**) Representative adult brains immunolabeled for *NPF*-specific antibody (red) and bruchpilot (grey) in *NPF-GAL4; nSyb-GAL80* flies with or without *UAS-NPF RNAi*. The scale bar corresponds to 50 μm. (**D**) Histogram showing Lb17 ♀-induced oviposition behavior in flies expressing *UAS-NPF RNAi* under *NPF-GAL4; nSyb-GAL80.* (**E**) Whole-mount preparations of mock and Lb17 ♀-exposed *NPF-GAL4; nSyb-GAL80 > UAS-NPF RNAi* ovaries that are immunostained with cDcp-1 (red) and DAPI (grey). The yellow arrowheads indicate the cDcp-1-positive egg chambers. The scale bar corresponds to 500 μm. Histogram showing (**F**) the average number of immature (light grey), as well as the matured eggs (green) and (**G**) the fold change in cDcp-1-expressing follicles in mock and parasitoid-exposed *NPF-GAL4; nSyb-GAL80 > UAS-NPF RNAi* flies. Error bars are ± SEM. ** $p \leq 0.01$ and *** $p \leq 0.001$ calculated using Student's *t*-test. For **D**: mock and Lb17-exposed egg-lay responses are presented as light and dark grey bars, respectively along with mean ± SEM. Refer to **Table 1** for 'N' and *p* values.

Lb17-induced egg-lay decrease (Fig 7A). The ovary expression of NPFR is much lower compared to the brain and the midgut cells [45,48,54], raising the possibility that very subtle changes in egg-laying may have been undetectable and/or not significant. These results

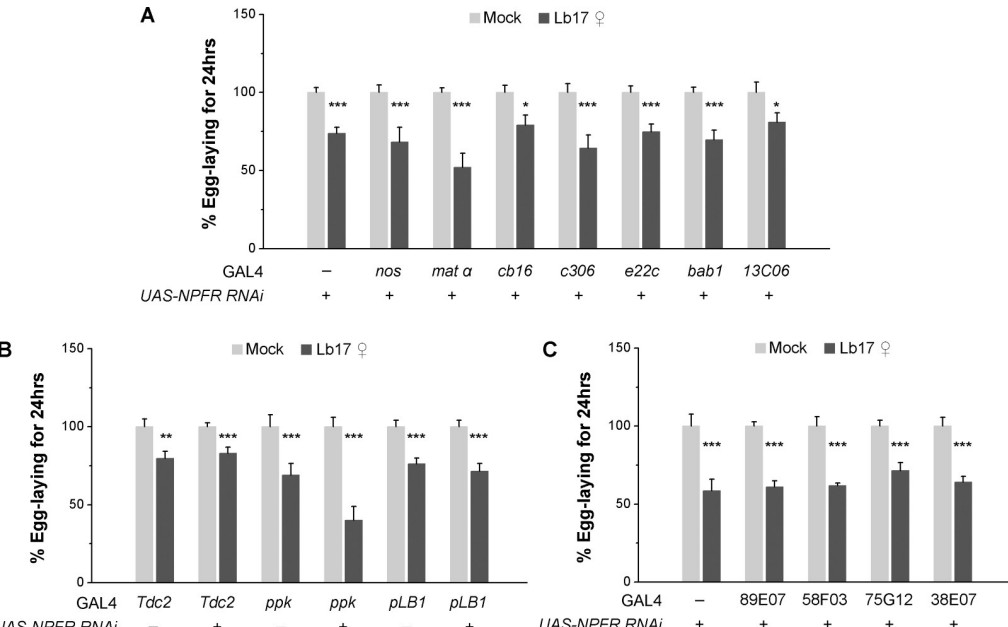

**Fig 7. Ovarian NPFR activity is dispensable for egg-lay reduction.** RNAi-mediated knockdown of NPFR in (**A**) a subset of germline cells, (**B**) *Tdc2*-, *ppk*- or *pLB1*-positive cells, and (**C**) GAL4 drivers expressed in various regions of the fan-shaped body has no effect Lb17-induced egg-lay responses. Light grey and dark grey bars respectively correspond to egg-lay responses of mock and Lb17 ♀-exposed flies. Error bars are ± SEM. * $p \leq 0.05$, ** $p \leq 0.01$ and *** $p \leq 0.001$ calculated using Student's *t*-test. Refer to **Table 1** for 'N' and *p* values.

demonstrate that the neuroendocrine signaling mechanism that regulates egg-lay depression may not be mediated through the ovarian NPFR. It also evident from the conclusion that Lb17-induced oviposition responses necessitate the CNS-expressed NPF, but not the hormonal NPF from the midgut EECs that targets non-neuronal NPFR (Fig 6).

To further investigate the subset of NPFR expressing cell-types involved in egg-lay depression, we tested the consequence of NPFR removal in previously identified neuronal subsets implicated in egg-laying behavior [53,56,57]. Female flies with NPFR knockdown in *Tdc2*-marked octopaminergic neurons (*Tdc2-GAL4*) [58] or a pair of oviduct-born *pickpocket* neurons (*ppk-GAL4*) [56,59] or in pLB1 cells that mediate egg-lay decrease after bacterial infection (*pLB1-GAL4*) [57] did not affect the egg-lay reduction observed after 24 hrs of wasp exposure (Fig 7B). Later, NPFR knockdown using GAL4 drivers that mark different regions of the fan-shaped body, a substructure of *Drosophila* central complex, whose function is implicated in inter-species communication [60] failed to suppress Lb17-induced behavioral modifications (Fig 7C). A more systematic dissection of NPFR-expressing cell-types further supported these observations (Fig 8), which suggests that the above-mentioned NPFR-positive cells are unlikely to be involved in parasitoid-generated innate behavioral responses.

Subsequently, we carried out a genetic screen using *NPFR-GAL4* drivers generated from the Fly Light project [61,62] and identified the NPF-responsive cell-types that are involved in the egg-lay depression (Fig 8). Compared to previously described *NPFR-GAL4* driver [49], the Fly Light *NPFR*-GAL4 lines showed much-restricted reporter expression in the brain (Fig 8A–8D) (http://flweb.janelia.org/cgi-bin/flew.cgi). Additionally, two out of four *NPFR-GAL4* drivers (60E02 and 65C12) showed expression in the ovaries (S5 Fig). Genetic perturbations using either an RNAi against NPFR or *UAS-Kir2.1* in NPFR-expressing cell-types that are marked by 60G05-GAL4 phenocopied the *NPFR* mutant behavior for Lb17-induced egg-lay depression (Fig 8B, 8F and 8J and Table 1). However, female flies expressing either *UAS-NPFR RNAi* or

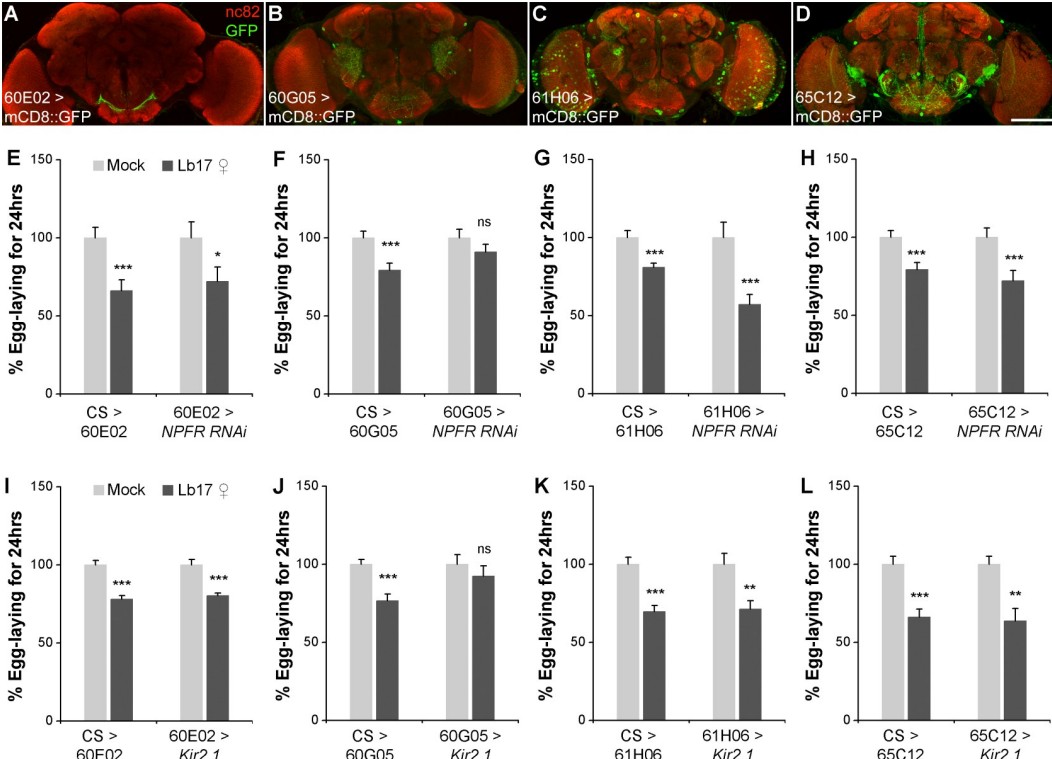

**Fig 8. Parasitoid-induced egg-lay depression is regulated by NPF-NPFR signaling.** (**A**) to (**D**) CNS expression of various *NPFR-GAL4* lines that are generated from the FlyLight project: (**A**) 60E02, (**B**) 60G05, (**C**) 61H06 and (**D**) 65C12-GAL4s driving *UAS-mCD8::GFP* reporter. Brains are immunostained with anti-GFP (green) and anti-bruchpilot (red). The scale bar corresponds to 100 μm. (**E**) to (**L**) Histograms showing Lb17 ♀-induced egg-lay responses of flies expressing either (**E-H**) an RNAi-against NPF receptor (*UAS-NPFR RNAi*) or (**I-L**) an inward-rectifying potassium ion channel (*UAS-Kir2.1*) in a subset of *NPFR*-positive cell-types. Mock and Lb17 ♀-exposed egg-lay responses are respectively represented as light and dark grey bars, along with mean ± SEM. * $p \leq 0.05$, ** $p \leq 0.01$, *** $p \leq 0.001$ and ns for non-significance ($p > 0.05$) calculated using Student's *t*-test. Refer to **Table 1** for 'N' and *p* values.

*UAS-Kir2.1* under 60E02-GAL4 (Fig 8A, 8E and 8I and Table 1), 61H06-GAL4 (Fig 8C, 8G and 8K and Table 1) and 65C12-GAL4 (Fig 8D, 8H and 8L and Table 1) appear to have no significant effect on Lb17-induced behavioral response. Additionally, knockdown of NPFR in 60G05-marked cell-types inhibits the retention of mature stage 14 follicles and blocks stage-specific apoptosis in the wasp-exposed female ovaries (Fig 9B and 9C). The absence of 60G05-GAL4 driven reporter expression in the ovaries (S5B Fig) further supports our conclusion that NPFR activity in the ovarian cells may be dispensable for the observed germline modifications. Together, our findings suggest that the brain-derived NPF recruits NPFR-expressing 60G05-positive cell-types that controls the Lb17-specific egg-lay depression. 60G05-GAL4 expressed both in the CNS and the VNS (Fig 9A), and our attempts to further dissect the cell-types that function to regulate the oviposition suppression failed due to lack of reagents that probe NPFR expression directly or GAL4 lines that dissect expression. Thus, we cannot exclude the possibility that 60G05-GAL4 driven NPFR depletion perturbs ovarian NPFR expression in an undetectable manner and/or alters NPFR expression in cells that are yet to be identified.

## Discussion

In the current study, we show that *Drosophila* females lay fewer eggs in the presence of a larval parasitoid *L. boulardi*. In addition to increased apoptosis of the vitellogenic follicles,

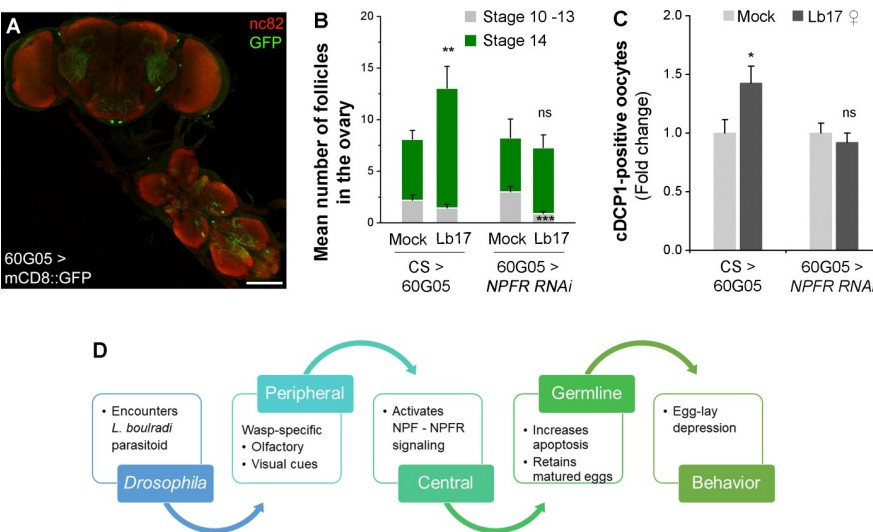

**Fig 9. A subset of NFPR-expressing cells are recruited by NPF signaling. (A)** Expression pattern of the 60G05-GAL4 driver in the brain and the ventral nerve cord. Preparations are immunostained with anti-GFP (green) and anti-bruchpilot (red). The scale bar corresponds to 100 μm. RNAi-mediated knockdown of NPFR using 60G05-GAL4 driver blocks Lb17-induced (**B**) matured follicles stalling in the ovaries and (**C**) apoptosis of the early egg chambers. Error bars are ± SEM. $^*$ $p \leq 0.05$, $^{**}$ $p \leq 0.01$, $^{***}$ $p \leq 0.001$ and ns for non-significance ($p > 0.05$) calculated using Student's $t$-test. (**D**) A model summarizing the neuroendocrine regulation of Lb17-induced egg-lay depression in *Drosophila*.

Lb17-exposed females transiently retain the matured egg chambers in their ovaries, together causing a significant reduction in egg-lay. These modifications in innate behavioral responses and associated germline physiology require both visual as well as parasitoid-specific odor perception (Fig 9D). Exposure to *L. heterotoma* (Lh14) also induces similar oviposition responses in fruit fly [16]. However, visual perception of Lh14 parasitoids by *Drosophila* females increases cell-death of the vitellogenic follicles that leads to long-term suppression of egg-laying, suggesting that although the final effect is the same, the underlying mechanisms of how *Drosophila* respond to each parasitic wasp are different. Given the variation in host range, immune suppression, and virulence strategies of these two *Leptopilina* wasp species [8–10], the observation of differences in the mechanisms that elicit oviposition behavior is not surprising. As mentioned above, the generalist *L. heterotoma* infects diverse species of *Drosophila*, whereas the specialist *L. boulardi* mostly parasitizes *D. melanogaster* and *D. simulans* clade [10]. The venom of *L. heterotoma* directly interferes with the host encapsulation response by attacking circulating lamellocytes [63,64]. In contrast, *L. boulardi* venom moderately alters host lamellocytes without lysing, thereby suppress the melanotic capsule formation [63,64]. Given the additional possible differences in the predation strategies of *Leptopilina* wasp species, we speculate that the fruit flies might have evolved both general anti-predation strategies as well as mechanisms that are unique to each predator.

In general, to overcome the stronger parasitic pressure imposed by *Leptopilina* wasps, the adult fruit flies have adapted egg-lay suppression behavior. First, in the presence of a *Leptopilina* wasp, *Drosophila* females retain the eggs and subsequently lay them in non-infested or protected oviposition sites. Our data demonstrate that upon wasp removal (Fig 1B–post-exposed), the oviposition behavior of Lb17-exposed females comparable to unexposed controls, indicating that the matured follicles are stockpiled and poised for oviposition. Retention of viable matured egg chambers in *Drosophila* is a strategy that has evolved for coping with different environmental stresses. For example, in the event of bacterial infection, *Drosophila*

females display a transient decrease in egg-laying that protects their progeny from similar infection. Bacterial peptidoglycan activates the NF-kB pathway in a subset of octopaminergic neurons that block egg release into the oviduct leading to the accumulation of matured follicles in the infected females, which are released after subsidization of bacterial peptidoglycan signaling [57]. Likewise, to resist environmental stresses such as starvation, low temperatures, and short photoperiods, *Drosophila* females switch to reproductive dormancy and hence reallocate resources to survival rather than reproduction [65]. By blocking the energy-consuming egg production, parasitoid-exposed females could redirect the reproductive resources to produce a smaller number of offspring with enhanced immunity and stronger resistance to parasitic infection. A most recent study showed that prolonged exposure of the parental flies to *L. heterotoma* wasps leads to visual cue-mediated depression of NPF-signaling in the adult fly brain. This triggers caspase-mediated germline apoptosis that not only reduces egg-lay but also accounts for epigenetic reprogramming in female germline and predisposition of ethanol preferences in the offspring that last up to five generations [5]. However, the underlying mechanisms in terms of neuronal circuits and molecules involved in behavioral and germline physiological modifications remain unclear.

The alteration of oviposition response to *Leptopilina* suggests that the neuronal circuit(s) that produce *Leptopilina* species-specific behavior must identify and distinguish between closely related wasp species. As discussed earlier, different sensory modalities such as olfactory [14–16], visual [15–17], and nociceptive cues [12,13] play a crucial role in recognition and avoidance of the predatory threat. For instance, except for iridoid-producing *Leptopilina*, neither larvae nor ovipositing adult females show ab10B ORN-mediated avoidance behavior towards other parasitoids [14]. In nature, adult flies encounter many types of larval parasitoids [8,9]. We, therefore, speculate that the fruit flies have evolved more sophisticated means for detecting larval parasitoids that later elaborated to give rise to parasitoid-specific responses. We note that apart from the egg-laying behavior reported here, adult *Drosophila* defensive strategies in response to pupal parasitoids remain unexplored.

We show that neuroendocrine signaling involving NPF and NPFR, which regulate various stress responses, are also required for parasitoid-induced germline modifications (Fig 9D). The recruitment of NPF-NPFR signaling requires parasitoid-specific visual and olfactory cues, and the NPF-responsive higher brain region that integrates these sensory inputs is yet to be identified. Reduced NPF expression in the *L. heterotoma* wasp-exposed fly brain functions to modify oviposition preferences [5,17], whereas mating-dependent increase in the mid-gut expression of NPF controls GSCs proliferation in the ovaries [45,52]. How do parasitoid-specific sensory modalities integrate and activate neuroendocrine signaling? How does NPF recruit a subset of NPFR-expressing cells that regulate germline physiology? Many questions remain open. Given the conserved function of NPF and its mammalian ortholog NPY in regulating stress responses [66–68], these questions are of fundamental relevance to both behavioral immunity and neuromodulation of germline physiology. These observations further support an ever-increasing body of evidence indicating the peripheral and CNS signaling via neuropeptides that regulate the GSC proliferation and germline development (Fig 9D). It will be of great interest for future experiments to dissect the precise mechanisms through which germline cells respond to neuronal inputs, and in turn, whether germline cells signal back to the brain to modify behavior. Finally, during normal physiological and pathological conditions local and/or distant neurotransmitter signaling such as GABA, dopamine, glutamate, serotonin, and NPY, regulate the proliferation and differentiation of adult neuronal stem cells in mammals [69–71]. Understanding whether the other cell populations including, a diverse population of adult stem cells, are continually modulated by sensory signals would have broader implications on the adult brain functions and maintenance.

## Materials and methods

### *Drosophila* stocks

Flies were raised on a standard cornmeal medium at 25°C in 12:12 hrs light/dark (LD) cycle-controlled incubators and Canton S (CS) flies were used as wild-type controls for all the experiments unless otherwise specified. The following transgenic flies were obtained from different sources: *Orco[1]* (# 23129) [28], *ninaB[1]* (# 24776) [23,24], *Or49a-GAL4* (# 9985) [14], *UAS-Or49a RNAi* (# 64581), *Or85f-GAL4* (# 23136) [14], *UAS-Or85f RNAi* (# 63033), *Or56a-GAL4* (# 9988) [36], *UAS-Or56a RNAi* (# 64955) [36], *UAS-Kir2.1* (# 6596) [25], *UAS-TNTVIF* (# 28840), *UAS-TNTG* (# 28838) [29,30], *NPF-GAL4* (II) (# 25681) [47], *NPF-GAL4* (III) (# 25682) [47], *nSyb-GAL80* (# 79028) [49,53], *elav-GAL4[C155]* (# 458), *nSyb-GAL4* (51941) [53], *UAS-NPFR RNAi* (# 25939) [45], *cb16-GAL4* (# 6722) [72], *c306-GAL4* (# 3743) [73], *e22c-GAL4* (# 1973) [74], *bab1-GAL4* (# 6802) [75], *13C06-GAL4* (# 47860) [61,76], 89E07-GAL4 (# 40553), 58F03-GAL4 (# 39187), 75G12-GAL4 (# 39906), 38E07-GAL4 (# 50007), 60E02-GAL4 (# 39250), 60G05-GAL4 (# 39259), 61H06-GAL4 (# 39281) and 65C12-GAL4 (# 39348) [61,62] were obtained from Bloomington *Drosophila* Stock Center. *UAS-NPF RNAi* (KK 108772), *UAS-NPFR RNAi* (KK 107663 and GD 9605) [45] and *UAS-Orco RNAi* (KK 100825) were from the Vienna *Drosophila* Resource Center (VDRC). *Or83b-GAL4* [28], *MB247-GAL4* [39], *Tdc2-GAL4* [58], and *UAS-mCD8::GFP* from Mani Ramaswami, *ey-GAL4* from Yashi Ahmed, *nos-GAL4-VP16* and *mat ∝-GAL4-VP16* from Sharon E. Bickel, *ppk-GAL4* [56,59] and *pLB1-GAL4* [57] from Julien Royet and *Tk-gut-GAL4* [51,52] from Irene Miguel-Aliaga. As previously described, the mutants *NPF[SK1]*, *NPF[SK2]*, and *NPFR[SK8]* were generated in an *yw* background using CRISPR/Cas9 method [45,46].

### Wasp rearing

The larval parasitoid, *Leptopilina boulardi* (Lb17 strain) and the pupal parasitoids, *Trichopria sp.1* (strain Trical) and *Pachycrepoideus sp.1* (strain Pac1Port) were provided by Todd Schlenke. To culture the parasitoid wasps, fruit flies (15 females and 5 males) were allowed to lay eggs for three days at room temperature (RT) in a standard *Drosophila* vial containing fresh media. For rearing Lb17 wasps, the adult fruit flies were replaced with adult Lb17 wasps (12 females and 5 males) for infecting the developing fly larvae. For culturing the pupal parasitoids, the adult wasps were introduced when the third instar *Drosophila* larvae initiate pupation. Lb17 and Trical parasitoids were maintained on *Drosophila melanogaster* (CS), whereas the Pac1Port parasitoids were raised on *Drosophila virilis*. To foster the parasitoid cultures, the vial plugs were supplemented with a 50:50 mixture of honey water. Unless otherwise mentioned, 5 to 10-day old female parasitoids were used for all the behavioral experiments.

### Egg-lay assays

To measure the egg-lay responses of parasitoid-exposed *Drosophila* females, newly eclosed 0 to 12 hrs old flies were aged 6 days at 25°C in 12:12 hrs LD cycle. After $CO_2$ anesthetization, 5 females and 2 male fruit flies along with 3 female parasitoids ('exposed' group) were transferred into a standard vial containing ~1 mL of fresh fly media. Age and genotype-matched control vials ('mock' group) were devoid of the parasitoids. Following 24 hrs of exposure at 25°C in 12:12 hrs LD cycle, flies were removed and the total number of eggs in each vial were counted manually using a ZEISS Stemi 2000 stereomicroscope. The egg-lay responses were represented as a percentage of the mock response of the flies (% egg-laying for 24 hrs). Mean egg-lay ± SEM values for all experimental genotypes that were tested along with the number of sets (*n*) and *p* values are presented in Table 1.

Pre-exposed, wasp-exposed, and post-exposed mean egg-lay responses that respectively correspond to a 24 hrs time interval before, during, and after wasp exposure for the same cohort of flies (Fig 1B). After $CO_2$ anesthetization, 6-day old experimental flies (5 ♀ and 2 ♂ flies per vial) were transferred into a fresh vial for 24 hrs the pre-exposed interval, then transferred to a fresh food vial containing a wasp for 24 hrs—a wasp-exposed interval, and finally, wasps were removed, and flies were transferred to a fresh food vial for 24 hrs—the post-exposed interval. The number of eggs was documented for every 24 hrs time interval (Fig 1B), and the eclosion rate was analyzed for all the time intervals (S1A Fig).

## Immunostaining

Briefly, adult brains were dissected [77] in 1X phosphate-buffered saline (PBS) and fixed in 4% paraformaldehyde (PFA) (Thermo Scientific, # 28908) in PBS for 30 min at RT. The samples were permeabilized with 0.3% PTX (0.3% of Triton X-100 in PBS) and incubated with primary antibodies for 48 hrs at 4˚C. After washings, the samples were labeled overnight at 4˚C with secondary antibodies. Thus, immunostained samples were rinsed with 0.3% PTX and mounted in Vectashield (H-1000, Vector Laboratories, CA).

For midgut preparations [78], the dissected samples were fixed in 4% PFA for 2 hrs at RT. Fixed samples were then washed with 0.1% PTX for 1 hr at RT and labeled overnight with primary antibodies at 4˚C. After 1 hr of washings, the samples were incubated with secondary antibodies for 2 hrs at RT. Before mounted in Vectashield the samples were washed again and stained with DAPI.

Primary antibodies used in this study were mouse anti-bruchpilot (nc82, 1:20) from Erich Buchner (University of Würzburg, Germany) and chicken anti-GFP (1:1000) from abcam (# ab13970). The custom-made rabbit polyclonal NPF antibody was generated against the NPF peptide sequence: C-Ahx-SNSRPPRKNDVNTMADAYKFLQDLDTYYGDRARVRFamide (21st Century Biochemicals, Marlboro, MA) and used at 1:2000 dilution [5]. Fluorophore-conjugated (Alexa Fluor 488 or 546 or 633) secondary antibodies from Invitrogen were used at 1:400 dilution. 1024 X 1024 pixels with 1 μm interval images were acquired using a Nikon A1R scanning confocal microscope (Melville, NY).

## Ovary preparations

Ovaries dissected (in 1X PBS) from mock and parasitoid-exposed *Drosophila* females (7-day old) were then gently transferred on a clean glass slide containing a drop of PBS. Whole-mount ovary images were captured using a Leica *MZ FL III* stereomicroscope with Olympus DP software.

For quantifying the number of GSCs and the apoptotic follicles, the tissue preparation was performed as previously described [79]. Briefly, the ovaries dissected in 1X PBS were fixed in 4% PFA for 15 min at RT, permeabilized with 0.2% PTX and labeled overnight with primary antibody at 4˚C (rabbit cleaved anti-Dcp-1, 1:200, Cell Signaling Technology, # 9578; rat anti-Vasa, 1:1000, Developmental Studies Hybridoma Bank (DSHB) and mouse anti-Hts 1B1, 1:50, DSHB). After washings, ovaries were incubated for 2 hrs at RT with fluorophore-conjugated secondary antibodies (Alexa Fluor 488 or 568), used at 1:400 dilution. The samples were then rinsed with 0.2% PTX, stained with DAPI for 10 min and mounted in Vectashield. Samples were visualized using a Nikon A1R confocal microscope or Nikon Eclipse E800 microscope. Based on the morphology and position of spherical spectrosome, GSC numbers were established [79]. Using a ZEISS Stemi 2000 stereomicroscope, we identified stage-10 to 14 egg chambers as previously described [22,55] and quantified the number of stage-10 to 13 and stage 14 follicles per ovary. No less than 10 ovaries were investigated per condition.

Images were processed using the Nikon NIS-Elements imaging software and Adobe Photoshop CS3 software (Adobe Systems Incorporated, USA).

## Statistical analysis

The OriginPro software (OriginLab) was used to plot the graphs and to perform the statistical tests. To calculate the differences between group means, *p* values were determined using two-sample unequal variance Student's *t*-test with 0.05 as the statistical significance level. Throughout this manuscript, the *p*-value is provided in comparison with mock and indicated as * $p \leq 0.05$, ** $p \leq 0.01$, *** $p \leq 0.001$ and ns for non-significance ($p > 0.05$).

## Supporting information

**S1 Fig.** (**A** and **A'**) Stacked histogram showing the average number of female (cyan) and male (light grey) flies eclosed from mock and wasp-exposed groups: (**A**) normalized and (**A'**) raw eclosion data. (**B**) and (**C**) Histogram showing egg-lay responses of wildtype Oregon R females to larval (Lb17) and pupal (Pac1Port and Trical) parasitoids. Light grey bars correspond to egg-lay responses of mock-exposed flies. Error bars are ± SEM. *** $p \leq 0.001$ and ns for non-significance ($p > 0.05$) calculated using Student's *t*-test.
(TIF)

**S2 Fig.** (**A**) Whole-mount preparations of mock and Lb17 ♀-exposed CS ovaries that are immunostained with DAPI (blue) and cDcp-1 (red). The arrowheads indicate the cDcp-1-positive egg chambers. The scale bar corresponds to 500 μm. (**B**) Stacked histogram showing the average number of stages 10–13 (light grey) and stage 14 (green) follicles in Oregon R female ovaries that are either exposed or unexposed to Lb17 ♀ wasps. Error bars are ± SEM. *** $p \leq 0.001$ and ns for non-significance ($p > 0.05$) calculated using Student's *t*-test.
(TIF)

**S3 Fig.** (**A**) Representative ovary images of CS flies that are exposed to Lb17 ♀ in the absence of light. Histogram showing the Lb17 ♀-induced oviposition behavior in flies expressing (**B**) *UAS-Orco RNAi* and (**C**) *UAS-Kir2.1* in ORNs. Error bars are ± SEM. *** $p \leq 0.001$ and ns for non-significance ($p > 0.05$) calculated using Student's *t*-test.
(TIF)

**S4 Fig.** (**A**) and (**B**) Confocal images of two different *NPF-GAL4* drivers expressing either (**A** and **B**) *UAS-mCD8::GFP* or (**A'** and **B'**) *UAS-NPF RNAi*. Brains are immunolabeled with anti-GFP (green), anti-NPF (red) and anti-bruchpilot (blue). The scale bar corresponds to 100 μm. (**C**) The average number of *NPF*-positive midgut cells in LB17 ♀-exposed (dark grey) CS flies are indistinguishable from their mock controls (light grey). (**D** and **E**) *tk-gut-GAL4 > UAS-NPF RNAi* flies showed significant reduction of the number of NPF-positive cells in the midgut. (**F**) Histogram showing the average basal egg-lay in flies expressing *UAS-NPF RNAi* using GAL4 drivers–*tk-gut-GAL4*, *elav-GAL4*, and *nSyb-GAL4*. Error bars are ± SEM. *** $p \leq 0.001$ and ns for non-significance ($p > 0.05$) calculated using Student's *t*-test.
(TIF)

**S5 Fig.** (**A-D**) Representative images of anti-GFP immunostaining in the ovaries of *NPFR-GAL4* drives: (**A**) 60E02, (**B**) 60G05, (**C**) 61H06 and (**D**) 65C12-GAL4s expressing *UAS-mCD8::GFP* reporter. Nuclei are stained with DAPI. The scale bar corresponds to 500 μm.
(TIF)

## Acknowledgments

We acknowledge Todd Schlenke for providing the wasp strains and Mani Ramaswami, Yashi Ahmed, Julien Royet, Irene Miguel-Aliaga, Sharon E. Bickel and the *Drosophila* stock centers for fly lines and Erich Buchner for nc82 antibody. We thank our laboratory members for useful discussions and the Dartmouth Department of Biological Sciences Light Microscopy Facility. Our special thanks to Katie A. Edwards and Victoria L. Marlar for maintaining the wasp and fly stocks during COVID-19 pandemic.

## Author Contributions

**Conceptualization:** Madhumala K. Sadanandappa, Giovanni Bosco.

**Data curation:** Madhumala K. Sadanandappa.

**Formal analysis:** Madhumala K. Sadanandappa.

**Funding acquisition:** Madhumala K. Sadanandappa, Giovanni Bosco.

**Investigation:** Madhumala K. Sadanandappa, Shivaprasad H. Sathyanarayana.

**Methodology:** Madhumala K. Sadanandappa.

**Project administration:** Madhumala K. Sadanandappa.

**Resources:** Shu Kondo.

**Supervision:** Giovanni Bosco.

**Validation:** Madhumala K. Sadanandappa.

**Writing – original draft:** Madhumala K. Sadanandappa.

**Writing – review & editing:** Madhumala K. Sadanandappa, Shivaprasad H. Sathyanarayana, Giovanni Bosco.

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
