## [Decision Letter · Decision Letter 0]

13 Dec 2020

Dear Dr Bosco,

Thank you very much for submitting your Research Article entitled 'Neuropeptide F signaling regulates parasitoid-specific germline development and egg-laying in Drosophila' to PLOS Genetics.

The manuscript was fully evaluated at the editorial level and by independent peer reviewers. The reviewers appreciated the attention to an important problem, but raised some substantial concerns about the current manuscript. Based on the reviews, we will not be able to accept this version of the manuscript, but we would be willing to review a much-revised version. We cannot, of course, promise publication at that time.

Should you decide to revise the manuscript for further consideration here, your revisions should address the specific points made by each reviewer. We deem that his will require additional experiments to strengthen the evidence for the brain neuronal requirement of NPF, for example using a broad neuronal driver to knockdown NPF (see the major point of Reviewer #3 that suggests using nsyb-GAL4). We do not consider it essential to conduct new experiments to identify NPF target neurons (point 6 of Reviewer #2). In addition, point 2 of Reviewer #3 may be adequately addressed without further experimentation, by incorporating a caveat in the text about the MB247 driver not expressing in all mushroom body neurons. We will also require a detailed list of your responses to the review comments and a description of the changes you have made in the manuscript.

If you decide to revise the manuscript for further consideration at PLOS Genetics, please aim to resubmit within the next 60 days, unless it will take extra time to address the concerns of the reviewers, in which case we would appreciate an expected resubmission date by email to plosgenetics@plos.org.

[LINK]

We are sorry that we cannot be more positive about your manuscript at this stage. Please do not hesitate to contact us if you have any concerns or questions.

Yours sincerely,

Alex P Gould

Guest Editor

PLOS Genetics

Gregory P. Copenhaver

Editor-in-Chief

PLOS Genetics

Reviewer's Responses to Questions

**Comments to the Authors:**

Reviewer #1: Sadanandappa et al. present a very interesting study combining Drosophila genetics with the naturally occurring phenomenon of wasp parasites that attack Drosophila larvae in high numbers in the wild. Interestingly, while some larvae manage to escape the attack, adult female flies still try to avoid egg-laying when wasps are present. How this works is not well understood and is elucidated in the present study. More generally, the study identifies a neuroendocrine mechanism that links sensory perception not only with behavior, but also with cellular physiology of an internal organ. That this axis between brain and organ is extremely important becomes increasingly clear. Nevertheless, we still know very little about it. Thus, this study is important and will be of interest to the broad readership of PlosGenetics.

The authors show that adult flies recognize and react to a combination of visual and olfactory cues produced by the wasp, Leptopilina, leading to a reduction in egg-laying through two different mechanisms, namely increased apoptosis in the germline and retention of maturing eggs. This effect is induced by brain-derived NPF and NPF receptors in the nervous system. By contrast, NPF in the gut or receptors outside the nervous system appear to be redundant. The authors also show that the suppression of egg-laying is independent of the mushroom body and therefore, likely, memory is not required or induced through wasp encounters.

Finally, the authors speculate that sensory signals might more generally modulate cell proliferation in distant tissues.

I enjoyed reading the paper and would like to see it published in Plos Genetics. The data is very sound and of high quality. The authors present many genetic controls throughout the paper, which is rare and commendable. Nevertheless, I have a few suggestions that could further improve the manuscript.

1. The last statement in abstract and manuscript, e.g. “These observations raise the intriguing

possibility that animals employ sensory-driven neuroendocrine signaling to modulate the proliferation and development of cells of distant tissues.” is, in my opinion, a bit strong. While the authors show that sensory cues and NPF are required to induce germline and behavioral changes, how this works and how direct this is, remains unclear. I would recommend to remove this statement from the abstract.

2. The authors use MB247-Gal4 to probe the role of the mushroom body. MB247 is one of the older drivers and primarily labels the α/β and γ lobes and not all types of Kenyon cells equally. While I agree that the mushroom body is likely dispensable for the observed effects, I would like to see this experiment repeated with an additional, broader driver, e.g. MB10B-Gal4.

3. The authors refer to the very nice publication by Ameku et al., PlosBiology 2018 that used a previously published driver, TKg-Gal4, which is supposedly expressed only in the gut but not in the brain. Unfortunately, this driver is not specific to the gut as recently stated in an erratum (DOI: 10.1016/j.celrep.2020.02.011). While this finding has no direct consequence on the presented results by Sadanandappa et al., I would still suggest to clarify this throughout the manuscript whenever the study by Ameku et al. is cited.

Reviewer #2: My review is uploaded as an attachment

Reviewer #3: Summary

The ability of organisms to protect themselves from parasites is important for the survival of species. In nature, the vasp majority of Drosophila larvae are parasitized by wasp species, the most common being Leptopilina boulardi (a more specific parasite) and Leptopilina heterotoma (a generalist that infects many Drosophila species). Various behavior responses to evade these parasites have evolved in Drosophila, including larval rolling behavior or adult changes in egg laying and alcohol-seeking behavior for oviposition (both aimed presumably at reducing the number of future larvae exposed to the parasite). In this study, Sadanandappa et al. investigate the mechanisms underlying Drosophila melanogaster egg laying changes in response to Leptopilina parasitoids using a combination of genetic approaches and microscopy. The authors report that olfactory and visual cues are required for reduction of egg laying in response to Leptopilina, and that death of vitellogenic follicles and accumulation of stage 14, mature oocytes are part of the cellular mechanisms involved in reducing the number of eggs laid. They also report that Drosophila brain-derived neuropeptide F (NPF) signaling is required for the increased death of follicles, accumulation of mature oocytes, and reduction in egg laying in response to exposure to Leptopilina.

Critique

The manuscript is clearly written, the study addresses a fascinating area of biology, and many of the key conclusions are well supported by the data. For example, experiments using mutants and RNAi knockdown convincingly demonstrate a role for visual and olfactory cues, as well as for NPF signaling in the egg-lay response of Drosophila to Leptopilina. The data showing that NPFR is required in 60G05-expressing neurons (using both NPRF RNAi and UAS-Kir2.1) are strong. The data showing death of vitellogenic follicles in response to exposure to Leptopilina (and that it depends on NPF signaling) are also convincing. However, some of the data in the manuscript are less convincing. For example, there are concerns with the quantification of follicles of different stages per ovary, and also regarding the data on the specific source of NPF required for the response to Leptopilina (see below for details). Some other clarifications are also needed, as outlined below.

Major Points:

- In all egg count graphs, the authors should show the actual average number of eggs laid per female instead of setting the control to 100%. In figure S1A, it is unclear why the egg counts of mock exposed females go down; it would be helpful if the authors could comment on possible reasons.

- The authors should clarify in the methods what the time points are for pre-exposure and post-exposure egg counts. Similarly, it is not clear in lines 144-146 how the quantification for “mean egg-lay responses for 72 hrs” was done.

- In multiple graphs along the manuscript (Fig. 2E, 3B,E, 4B, 5E, 6F, 9B, S2B), the authors show quantification of the number of follicles at different stages of development per ovary. Given that it would not be trivial to do this precisely in whole ovaries, the authors should explain in detail how this was accurately done. Adding to this concern, the quantified numbers of stages 10-13 per ovary seem exceedingly low in these graphs. (Specifically, each ovariole under ideal conditions would be expected to have 1 or 2 of those stages, and each ovary has approximately 15 ovarioles, such that controls would be expected to have about 15-30 of those stages per ovary. Yet, in most graphs the control numbers are much lower - this is very puzzling.)

- In Fig. 2 and S2, the authors measure the number of GSCs, but the rationale for looking at GSC number (instead of proliferation, for example) over a 24-hour period is not clearly explained. Also, even if they examined proliferation, any changes in GSC proliferation would not translate into an immediate change in the number of eggs produced, since it takes close to 10 days from a GSC division to a mature egg. On a related note, in lines 188-190, the authors state: “These data indicate that 24 hrs of Leptopilina exposure appears to have no long-lasted effect on GSCs proliferation.” However, they did not measure proliferation of GSCs, they only measured GSC numbers, which are not affected by their proliferation rates (i.e. differences in GSC proliferation alter the number of cystoblasts produced, but not the number of GSCs, given that each division of a GSC generates one GSC for self-renewal and one cystoblast for differentiation). Related to that, in line 51, it might be more appropriate to refer to survival instead of proliferation.

- It would be interesting to know if NPF levels are altered in response to visual and olfactory inputs from Leptopilina exposure (e.g. by comparing NPF levels using antibodies used in the manuscript between wildtype versus genetically manipulated flies with impaired vision or olfaction exposed to Leptopilina or mock exposed.)

- Fig. S4D,E: the authors should clarify if the tk-gut driver is expressed in just a subset of enteroendocrine cells (i.e. this would help explain why so many NPF-positive cells are left in the gut after NPF knockdown using this driver).

- Fig. 6C and lines 343-346: there is a marked reduction in NPF levels even though the authors state they used NPF-Gal4 combined with nSyb-Gal80, which is puzzling. In any case, the fact that NPF levels are very reduced but the response to Leptopilina appears to be fully intact significantly weakens the authors’ conclusion that “neuronally-expressed NPF seems to mediate the Lb17 (…) responses.” (Could there be another non-neuronal, non-gut source of NPF?) The authors should use a neuronal driver (e.g. nSyb-Gal4) to knock down NPF in the brain to more conclusively test if NPF is indeed required in neurons for the response to Leptopilina.

- Lines 361-362 and Fig. 7a: standard UAS constructs are very poorly (if at all) expressed in the female germline, so the authors cannot conclusively rule out a possible requirement for NPFR in the germline based on those experiments. (UASp or UASz transgenes are required for effective expression in the female germline.)

- Lines 487-492: in the discussion, instead of just raising it as a possibility based on their study, the authors should briefly discuss existing evidence that distant neuronal inputs, including sensory inputs, can indeed influence mammalian neurogenesis (e.g. see reviews: Obernier & Alvarez-Buylla, Development 2019, and Ryu et al. Molecular Brain 9:43, 2016).

- in the Methods, more details should be including when explaining immunostaining procedures. Alternatively, reference citations could be included instead, if these procedures have already been explained in detailed previously.

Minor Points:

- instead of using “mid-oogenesis stages” through the manuscript, it would be more accurate to refer to those as vitellogenic follicles because more than half of the actual time of oogenesis (from GSC to stage 14) is actually spent in the germarium (e.g. see Lin & Spradling 1993, and Margolis & Spradling Development 1995).

- reference citations for genetic tools used in the manuscript are missing in multiple cases (e.g. lines 238-239; lines 280-281; line 334; line 374; also in methods).

- throughout graphs showing stages 10-13 and 14 quantification, Y-axis should refer to number of follicles (not eggs) in the ovary.

- there is a typo in Figure 9C (Fold-change).

- another suggested keyword: oogenesis

**Have all data underlying the figures and results presented in the manuscript been provided?**

Reviewer #1: Yes

Reviewer #2: **No: **At least I did not see any

Reviewer #3: Yes

PLOS authors have the option to publish the peer review history of their article (what does this mean?). If published, this will include your full peer review and any attached files.

Reviewer #1: No

Reviewer #2: No

Reviewer #3: No

---

## [Decision Letter · Decision Letter 1]

1 Mar 2021

Dear Dr Bosco,

We are pleased to inform you that your manuscript entitled "Neuropeptide F signaling regulates parasitoid-specific germline development and egg-laying in Drosophila" has been editorially accepted for publication in PLOS Genetics. Congratulations!

Yours sincerely,

Alex P Gould

Guest Editor

PLOS Genetics

Gregory P. Copenhaver

Editor-in-Chief

PLOS Genetics

Comments from the reviewers (if applicable):

Reviewer's Responses to Questions

**Comments to the Authors:**

Reviewer #1: I thank the authors for addressing all of my comments and concerns. I am satisfied with their responses and can now recommend the manuscript for publication in Plus Genetics.

Reviewer #2: The authors addressed most of my queries in a satisfactory fashion. In the light of problems with lab work during the covid pandemic, I do not insist on the proposed experiments. The paper is ready to go!

Reviewer #3: Sadanandappa et al. have satisfactorily addressed my comments and I recommend publication of this revised manuscript in PLOS Genetics.

**Have all data underlying the figures and results presented in the manuscript been provided?**

Reviewer #1: None

Reviewer #2: None

Reviewer #3: Yes

PLOS authors have the option to publish the peer review history of their article (what does this mean?). If published, this will include your full peer review and any attached files.

Reviewer #1: No

Reviewer #2: No

Reviewer #3: No

**Data Deposition**

http://datadryad.org/submit?journalID=pgenetics&manu=PGENETICS-D-20-01660R1

**Press Queries**

---

## [Editor Report · Acceptance letter]

23 Mar 2021

PGENETICS-D-20-01660R1 

Neuropeptide F signaling regulates parasitoid-specific germline development and egg-laying in Drosophila 

Dear Dr Bosco, 

We are pleased to inform you that your manuscript entitled "Neuropeptide F signaling regulates parasitoid-specific germline development and egg-laying in Drosophila" has been formally accepted for publication in PLOS Genetics! Your manuscript is now with our production department and you will be notified of the publication date in due course.

With kind regards,

Andrea Szabo

PLOS Genetics

On behalf of:
